# Efficacy of Three Numerical Presentation Formats on Lay People’s Comprehension and Risk Perception of Fact Boxes—A Randomized Controlled Pilot Study

**DOI:** 10.3390/ijerph20032165

**Published:** 2023-01-25

**Authors:** Pascal Aubertin, Thomas Frese, Jürgen Kasper, Wilfried Mau, Gabriele Meyer, Rafael Mikolajczyk, Matthias Richter, Jan Schildmann, Anke Steckelberg

**Affiliations:** 1Institute of Health and Nursing Science, Interdisciplinary Center for Health Sciences, Martin Luther University, Halle-Wittenberg, 06112 Halle (Saale), Germany; 2Institute of General Medicine, Interdisciplinary Center for Health Sciences, Martin Luther University, Halle-Wittenberg, 06112 Halle (Saale), Germany; 3Department of Nursing and Health Promotion, Faculty of Health Sciences, Oslo Metropolitan University, 0176 Oslo, Norway; 4Institute of Rehabilitation Medicine, Interdisciplinary Center for Health Sciences, Martin Luther University, Halle-Wittenberg, 06112 Halle (Saale), Germany; 5Institute of Medical Epidemiology, Biometrics and Informatics, Interdisciplinary Center for Health Sciences, Martin Luther University, Halle-Wittenberg, 06112 Halle (Saale), Germany; 6Institute of Medical Sociology, Interdisciplinary Center for Health Sciences, Martin Luther University, Halle-Wittenberg, 06112 Halle (Saale), Germany; 7Institute for History and Ethics of Medicine, Interdisciplinary Center for Health Sciences, Martin Luther University, Halle-Wittenberg, 06112 Halle (Saale), Germany

**Keywords:** fact boxes, evidence-based health information, consumer health information, Patient Education Handout, presentation formats

## Abstract

(1) Background: Fact boxes present the benefits and harms of medical interventions in the form of tables. Some studies suggest that people with a lower level of education could profit more from graphic presentations. The objective of the study was to compare three different formats in fact boxes with regard to verbatim and gist knowledge in general and according to the educational background. (2) Methods: In May 2020, recruitment started for this randomized controlled trial. Participants were given one out of three presentation formats: natural frequencies, percentages, and graphic. We used Limesurvey^®^ to assess comprehension/risk perception as the primary outcome. The Kruskal–Wallis test and the Mann–Whitney U test were used in addition to descriptive analyses. (3) Results: A total of 227 people took part in the study. Results of the groups were nearly identical in relation to the primary outcome verbatim knowledge, likewise in gist knowledge. However, participants with lower educational qualifications differed from participants with higher educational qualifications in terms of verbatim knowledge in the group percentages. (4) Conclusions: The results indicate that all three forms of presentation are suitable for conveying the content. Further research should take the individual preferences regarding the format into account.

## 1. Introduction

In the past, the lack of evidence-based health information has been reported repeatedly [1,2]. However, members of the public have the right to be given evidence-based information [3] as a prerequisite for making informed choices about health-relevant issues. Fact boxes provide evidence-based health information and were also developed to present the benefits and harms of pharmaceutical products in an undistorted and comprehensible way. For this purpose, they are displayed as tables showing the benefits and harms, with or without additional text [4]. In Germany, the Harding Centre for Risk Literacy developed fact boxes on a wide range of health-related topics. They publish fact boxes indicating natural frequencies. Sometimes, the information is supplemented by pictograms [5]. The fact boxes are freely accessible on the internet. Fact boxes can support the patients and doctors in the process of shared decision making [4].

The authors of the guideline evidence-based health information conducted an evidence synthesis to present the effects of the fact boxes in comparison with other formats. The guideline could only include two RCTs that showed a positive effect of the fact boxes regarding the outcomes risk perception and knowledge, and comprehensibility and readability. As only two small studies were included, the guideline did not give a clear recommendation [4,6,7].

For this study, we updated the systematic literature search conducted for the guideline and found two more studies. Aiken et al. reported that readers prefer fact boxes to other formats, and that the recall of the risks of a medication improved in comparison to other formats [8]. Brick et al. showed that, in terms of knowledge and comprehension, the content of a fact box was better understood after six weeks compared to a text format [9].

However, there are differences in the way the tables are presented and in the accompanying texts. Data on the frequencies of benefits and harms are shown numerically (natural frequencies or percentages) or graphically in fact boxes [6,7,10,11]. The benefit of using graphics instead of numerical frequency information could not be shown in the evidence synthesis of the guideline evidence-based health information [4], but there are indications that groups with a low level of education could profit from the use of graphical presentations [4,12]. Other studies have, however, not been able to confirm these findings [4,13,14,15,16]. In a recent study, McDowell et al. showed that, regarding the outcome knowledge and understanding, there is no difference between the presentation in natural frequencies compared to a graphical presentation [17]. Due to the lack of evidence, it also remains questionable whether presentation as a percentage is equivalent in terms of knowledge or understanding compared to presentation as natural frequencies [4]. A preliminary study has shown that the natural frequency presentation was barely understood, and that the percentage presentation was preferred [18]. Users with a low level of education might benefit from the presentation of a bar chart [4,12].

To date, fact boxes have barely been evaluated [19]. 

The objective of the current pilot study was to compare the three formats natural frequencies, percentages, and bar graphs in fact boxes with regard to comprehension in the general population and in groups with varying educational background. We used antibiotics for acute bronchitis as an example. In addition, this study aimed to determine the feasibility and acceptability of the knowledge test as well as the necessary size of the sample in the planned study and the study procedures for the comprehensive randomized controlled trial (RCT).

## 2. Materials and Methods

The study is reported according to the Consolidated Standards of Reporting Trials (CONSORT) [20].

### 2.1. Design

This randomized controlled 3-arm pilot study was conducted as part of the “HeReCa-Online Panel” (Health Related Beliefs and Health Care Experiences in Germany), hosted by the Interdisciplinary Center for Health Sciences of the Martin Luther University Halle-Wittenberg.

### 2.2. Setting and Study Participants

This study included only people who were at least 18 years old at the time of the pilot study. A subsample was recruited among residents of Berlin, the capital city of Germany.

### 2.3. Recruitment

The current study recruited participants from the HeReCa panel (Health Related Beliefs and Health Care Experiences in Germany). This panel was initiated in 2019 by the Interdisciplinary Center for Health Sciences of the Martin Luther University Halle-Wittenberg and has already been used in a couple of studies related to health-related beliefs and health-care experiences.

Our study addressed the entire group of 241 new HeReCa panel members gained during the recent recruitment wave in Berlin. For this group, our study was the first study for these new panel members.

### 2.4. Randomization and Blinding

We used computer-generated random numbers for simple randomization of subjects. This feature was implemented in the Limesurvey^®^ platform, therefore ensuring allocation concealment. The participants were randomly allocated to one of the three groups and were not aware to which group they belonged.

### 2.5. Intervention and Control Intervention

The three fact boxes on the subject “Antibiotics for acute bronchitis” contained information concerning the benefits and harms of antibiotic treatment compared to treatment with a placebo or no treatment at all. The sources of the information were also provided. Participants randomly received one of the three formats: Figure 1: the fact box that presented treatment effects as natural frequencies; Figure 2: the fact box using percentages, and Figure 3: the fact box using a graphical presentation [21]. Bar charts were used for the graphical presentation of the frequencies, as recommended by Lühnen et al. [4]. Each group was given the same additional information as in the fact box (Figure 4).

### 2.6. Outcomes and Moderating Variables

Within the scope of this study, the outcomes “knowledge (verbatim and gist)”, “readability/comprehensibility”, “acceptance”, and “relevance”, which are established in the field of health communication, were collected using a questionnaire (see Table A1). The development of the questions on the primary outcome questions, as well as the questions on the secondary outcomes, was guided by the literature [6,7]. The knowledge questions and answers had been adapted from previous studies that surveyed knowledge for different topics. All these previous questionnaires were based on the General Medical Council’s ethical guidelines [22,23,24]. The questions to survey readability/comprehensibility, acceptance, and relevance were all taken from studies that were included in the guideline evidence-based health information [25]. The complete questionnaire was pre-tested in a previous focus group study through an iterative process, using the think-aloud method. The process was carried out until all questions were understandable [18].

Verbatim and gist knowledge

Verbatim knowledge was assessed using nine items and referred to the ability of the respondents to express correctly the absolute frequencies of the benefits and harms of antibiotic treatment, placebo treatment, or of no treatment at all. Each answer was coded as correct (1) or incorrect (0) and a sum score was calculated (see Table A1).

Furthermore, gist knowledge was surveyed to determine the extent to which respondents understood the importance of the differences between the benefits and harms of antibiotic treatment, placebo, or no treatment at all. Five multiple-choice questions were used here; the correct answers were coded with 1, the incorrect answers were coded with 0, and a sum score was calculated (see Table A1).

Readability/Comprehensibility

The readability and comprehensibility referred to how the recipients assessed the plausibility of the contents, texts, and tables [26]. The evaluation comprised four questions, using a four-point Likert scale (e.g., 3 = very understandable, 2 = understandable, 1 = incomprehensible, and 0 = absolutely incomprehensible) as an example (see Table A1).

Acceptance

Acceptance was defined as the attitude of the participants to the fact box. The evaluation comprised questions about the attractiveness of the design, the trustworthiness of the information, and the recommendation of the fact box using a four-stage interval scale (see Table A1).

Relevance

Relevance was defined as the assessment of whether the information was considered helpful. This outcome was surveyed by a single question, using a four-stage interval scale (3 = extremely helpful, 2 = helpful, 1 = hardly helpful, and 0 = not at all helpful) (see Table A1).

The questionnaire had been piloted in a previous focus group study [18].

Moderating variables

Based on the literature, the level of education was chosen as a moderating variable for the purpose of following sensitivity analysis. Participants with a lower secondary and a secondary school certificate were assigned to the group with a lower level of education. Participants with a higher school certificate or an academic degree were assigned to the group with a higher level of education.

### 2.7. Sample Size

A sample size calculation was not carried out for the pilot RCT [27].

### 2.8. Procedure of Data Collection

During enrolment into the study, the participants were asked to give written informed consent and to provide their email addresses so that they could be contacted for further surveys. In the next step, the participants were directed to the fact box and the corresponding questionnaire (implemented in Limesurvey^®^). Having read the fact box, the participants were asked to answer the questions about the outcomes under investigation. The fact box was accessible at all times while answering the questions. In addition, socio-demographic characteristics were collected.

The fact boxes survey was opened on 28 May 2020 and closed on 25 January 2021. No reminder letters were sent. No expense allowances were paid.

### 2.9. Data Analysis

The evaluation of the results comprised descriptive and inferential analyses. The statistical analyses for the primary outcome and the secondary outcomes were conducted using the Kruskal–Wallis test. The statistical analyses for knowledge according to educational level were conducted using the Mann–Whitney U test. The significance level for *p*-value was set at <0.05. The analyses were all performed as intention-to-treat. For the analyses, we used the software SPSS^®^ Version 27.0.1.0. In addition, data were split into two groups: a group with lower-level education and a group with higher-level education.

**Trial registration:** ISRCTN17033137: https://doi.org/10.1186/ISRCTN17033137. Date of registration: 06/04/2020.

## 3. Results

Out of the 8934 people invited, a total of 241 registered for the panel (2.7%) and 227 took part in the study (94.2%). More women than men participated in the study. A smaller number of people with a low level of education took part (secondary school or lower) than people with a high level of education (university admission and academic graduation) (Table 1).

All the participants were included in the evaluation (the details are shown in the CONSORT flow chart) (Figure 5).

### 3.1. Primary Outcome

#### Verbatim and Gist Knowledge

Regarding both verbatim knowledge and gist knowledge, the questions were answered correctly by the majority of the participants in each group. The primary outcomes were very similar across the groups (Table 2). With regard to verbatim knowledge, the percentage of missing answers in the groups was 18.3% (natural frequencies), 15.3% (percentages), and 17.8% (graphic chart). In relation to gist knowledge, the percentage of missing answers in the groups was 15.9% (natural frequencies), 12.5% (percentages), and 13.7% (chart).

### 3.2. Secondary Outcomes

#### 3.2.1. Readability/Comprehensibility

The majority of the participants in all three groups found all the fact boxes to be easily readable and comprehensible (Table 3). No differences were found. However, there were many missing answers on readability (natural frequencies group: 20.7%, percentages group: 18.1%, and graphic chart group: 23.3%).

#### 3.2.2. Acceptance

The acceptance for each of the fact boxes was high (Table 3) and the groups did not differ in their appraisal. With regard to acceptance, there were many missing answers (natural frequencies group: 18.3%, percentages group: 16.7% and graphic chart group: 23.3%).

#### 3.2.3. Relevance

The majority of the participants found the information to be helpful (Table 3). This response did not differ between the study groups. Almost one fifth of the answers on relevance were missing (natural frequencies group: 20.7%, percentages group: 18.1%, and graphic chart group: 23.3%).

### 3.3. Knowledge by Educational Level

The natural frequencies format showed comparable results between the two educational levels. In contrast, participants with a lower level of education who worked on the formats percentages and graphic chart achieved significantly lower mean scores with regard to verbatim knowledge. There were no differences with respect to gist knowledge (Table 4).

## 4. Discussion

The content of the fact boxes was well understood by the majority of the participants in all the groups, and the participants’ attitudes to the fact boxes were positive. The information provided by the fact boxes was seen as positive by most of the participants.

McDowell et al. [17] reported comparable results with regard to comprehension, as did the recently published study by Danya et al. [28]. McDowell et al. also showed a general but small long-term effect for improvement over 6 months. Comparable to our results, the sample was also well educated [17]. Danya et al. compared the column charts and pictograms [28].

Compared with other formats, fact boxes can substantially improve knowledge and risk perception [6]. Tait et al. investigated whether people with a lower level of education profit from the graphical presentation of the frequencies [12]. Our results showed that participants who worked on the natural frequencies and the graphic format achieved comparable results regardless of their educational level. There were differences in terms of verbatim knowledge between the format percentages grouped by educational level. However, the differences were rather small and likely not clinically relevant. Comparable results in terms of graphic charts have already been reported in other studies in the past [4,13,14,15,16]. The recommendation that natural frequencies, rather than percentages, should be used in communicating health information [29] particularly refers to conditional probabilities, for example, to estimate the predictive values of diagnostic tests [30]. Conclusiveness of the results of the current study is limited in some respects. Firstly, the current study used the topic “Antibiotics for acute bronchitis” as an example to compare the three fact box formats [21]. It is unclear whether the results are generalizable to other medical topics or fact boxes using alternative graphical presentation formats instead of bar charts. Secondly, the results have been gained on an aggregated level, revealing the lack of advantage of one of the fact box formats presented in mean values over the total group. This approach, however, is not sensitive to preferences and advantages of formats on the level of subgroups; if existing, the latter are likely to disappear in the overall examination. More research is needed to better understand variables and processes allowing for the meaningful tailoring of risk information on the individual or group level. Research to gain this knowledge also needs to consider the role preferences for the choice of respective formats. In the current study, the users did not have any choice. Therefore, we do not know whether the participants had been assigned to a group where their own preferred format was the subject in question. Self-assignment to a risk communication format might lead to better and more divergent results [31]. The redevelopment of the PREDICT tool that has been developed to predict survival in breast cancer now allows users to choose between different formats [32]. However, systematic evaluation regarding the effects on knowledge and decision making is pending. Likewise, publishers of fact boxes should provide the target groups with varying formats of presentation of the frequency information in fact boxes.

### Strengths and Limitations

The pilot study was conducted online which had the advantages of reaching the participants quickly and the answers being promptly available in a digital format. A further strength of the study can be seen in blinding, randomization, and intention-to-treat analysis since these criteria demonstrate a high quality of evidence [33]. Among those who registered for the panel, the participation in this study was very high.

In the study, only the example “Antibiotics for acute bronchitis” from the Martin Luther University fact box was used [20]. It is unclear whether the results would be similar if alternative graphical presentations were used instead of bar charts.

All outcomes were obtained immediately after reading the fact boxes. It is not known whether knowledge would decrease in follow-up assessments.

Participants with a low level of education were underrepresented in this study. The breaking point between the two education groups had been defined arbitrarily. Shifting the breaking point could produce different results. In addition, participants were assigned using randomization to presentation formats that were possibly not matching their individual preferences. This could have impacted their performances and might have biased the results.

A limitation of the online survey was the fact that people without an internet connection and/or email address and/or without an internet-enabled device (terminal) were not able to take part in the study. It must also be assumed that older citizens could only be minimally represented by an online survey, as only half of those over 70 use the internet [34].

## 5. Conclusions

The application of fact boxes around the time when information is needed is beneficial but the knowledge of patients receiving information in the three formats did not seem to differ. However, level of education might make a difference. In addition, further research should keep the individual preferences in mind and offer different formats for choice.

## Figures and Tables

**Figure 1 ijerph-20-02165-f001:**
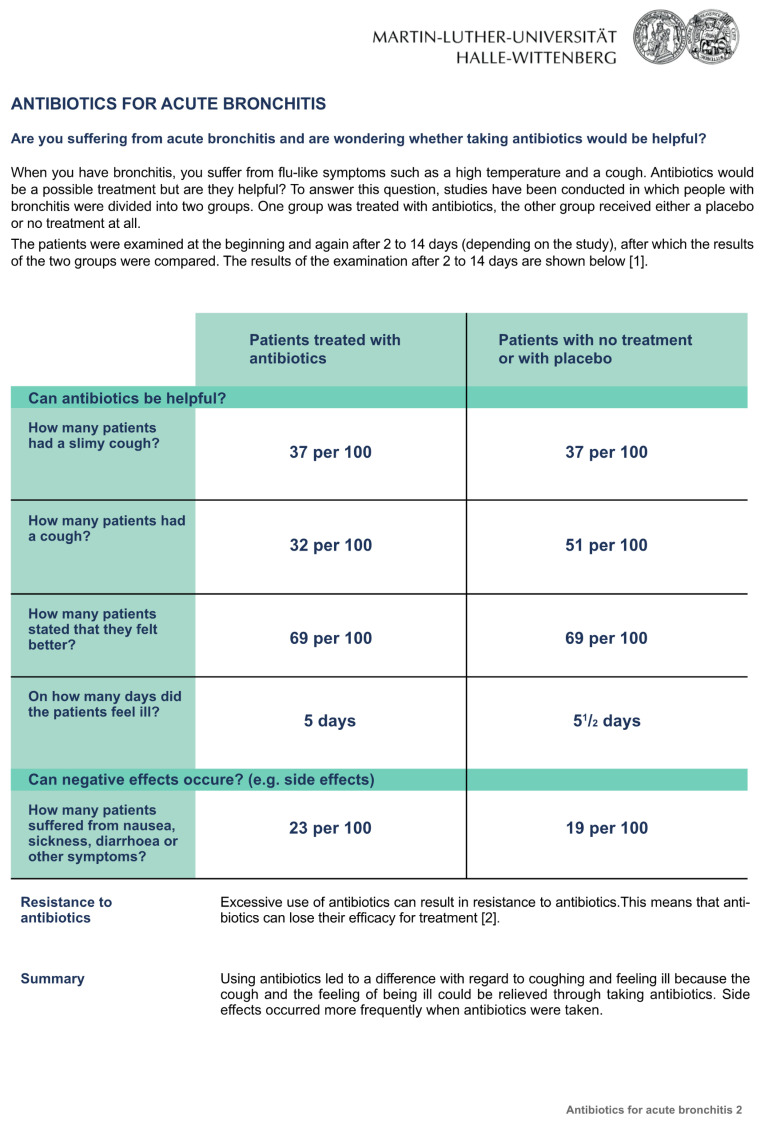
Fact box that presented treatment effects as natural frequencies.

**Figure 2 ijerph-20-02165-f002:**
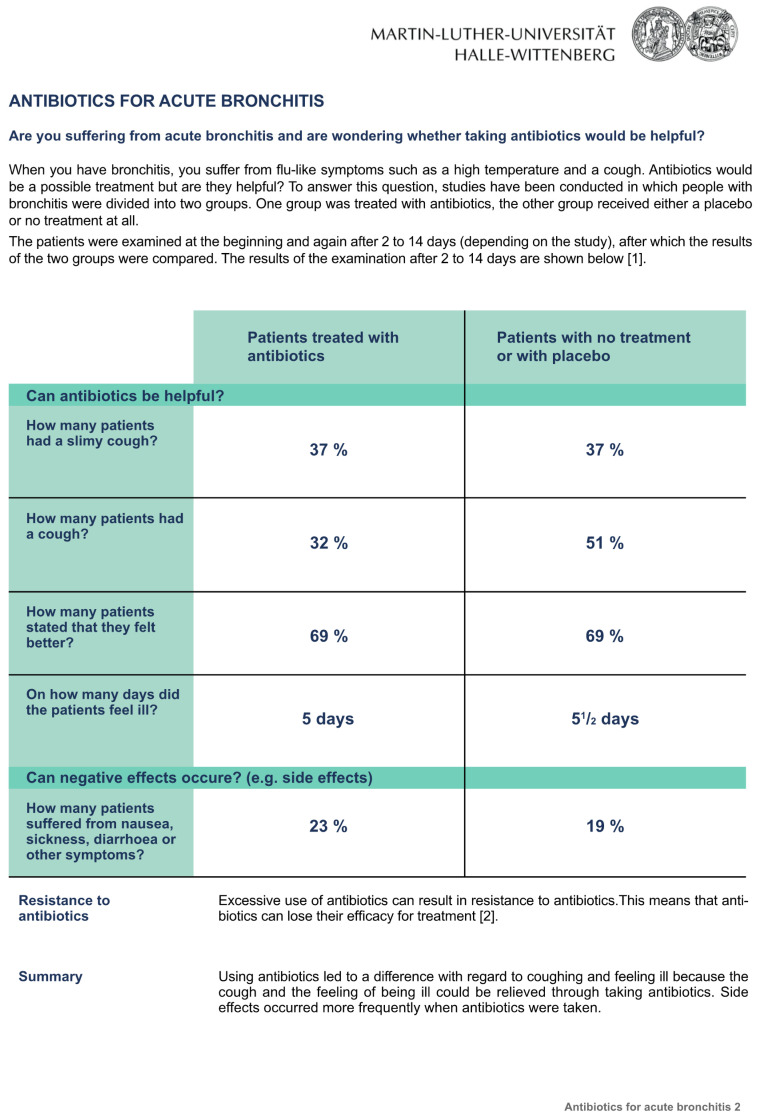
The fact box that presented treatment effects as percentages.

**Figure 3 ijerph-20-02165-f003:**
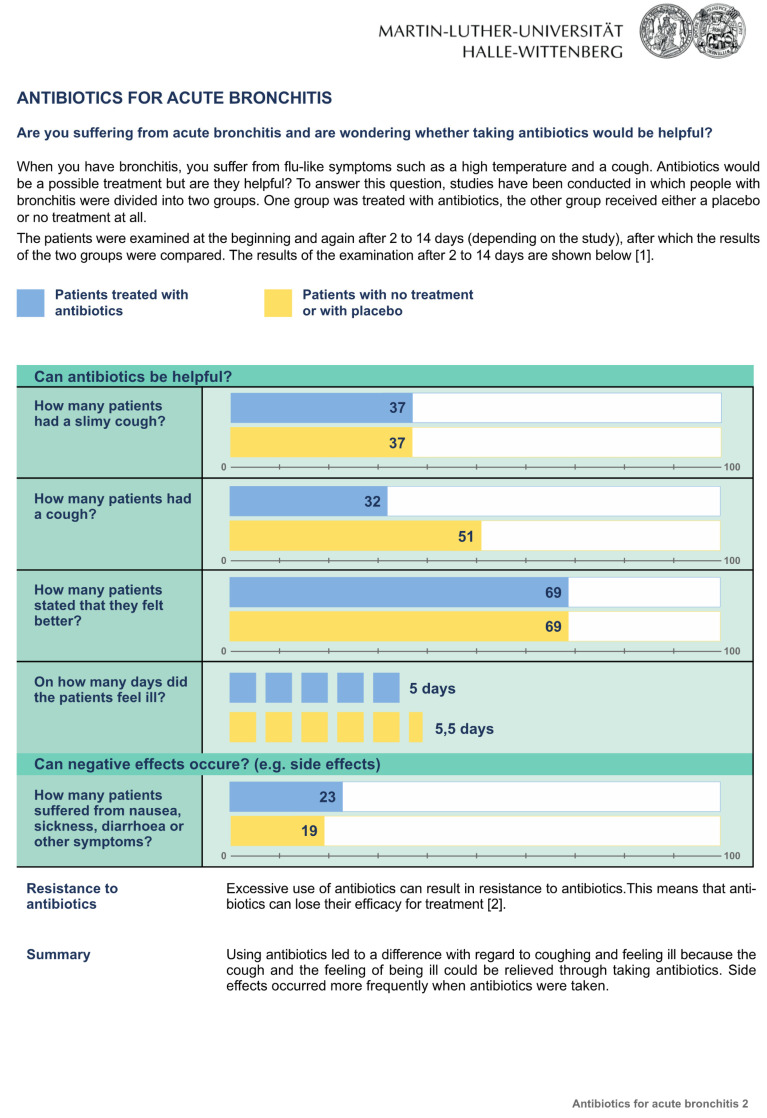
The fact box that presented treatment effects as a graphical presentation.

**Figure 4 ijerph-20-02165-f004:**
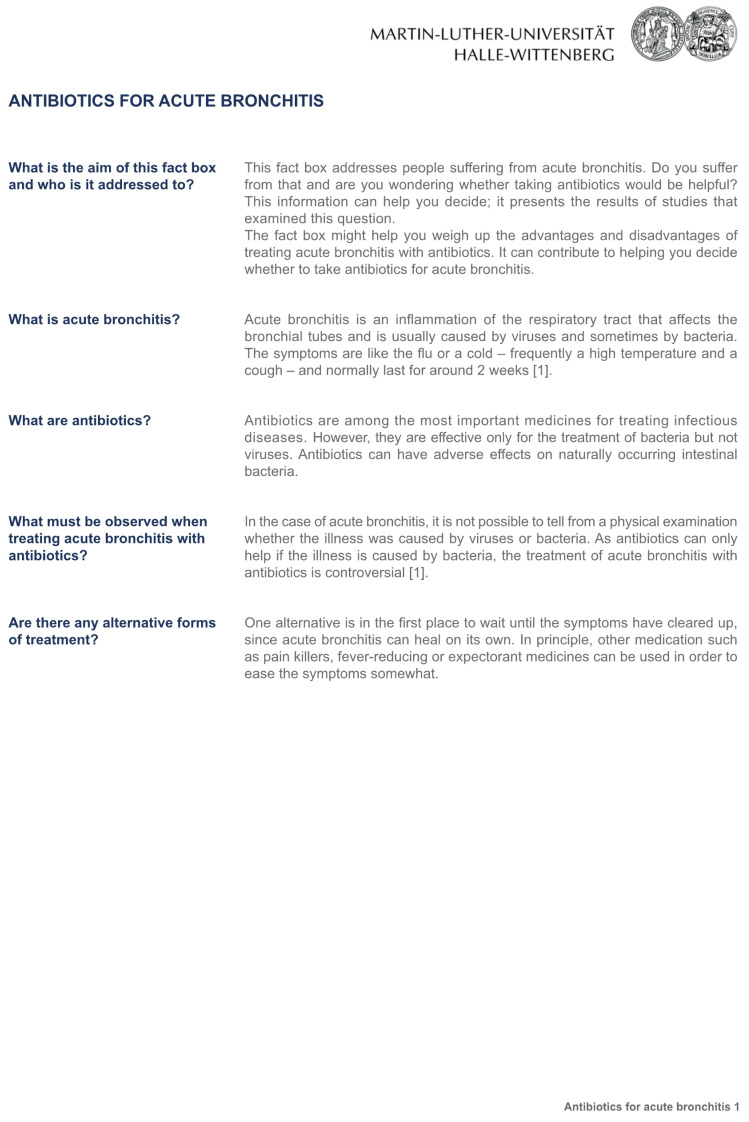
Fact boxes with additional information.

**Figure 5 ijerph-20-02165-f005:**
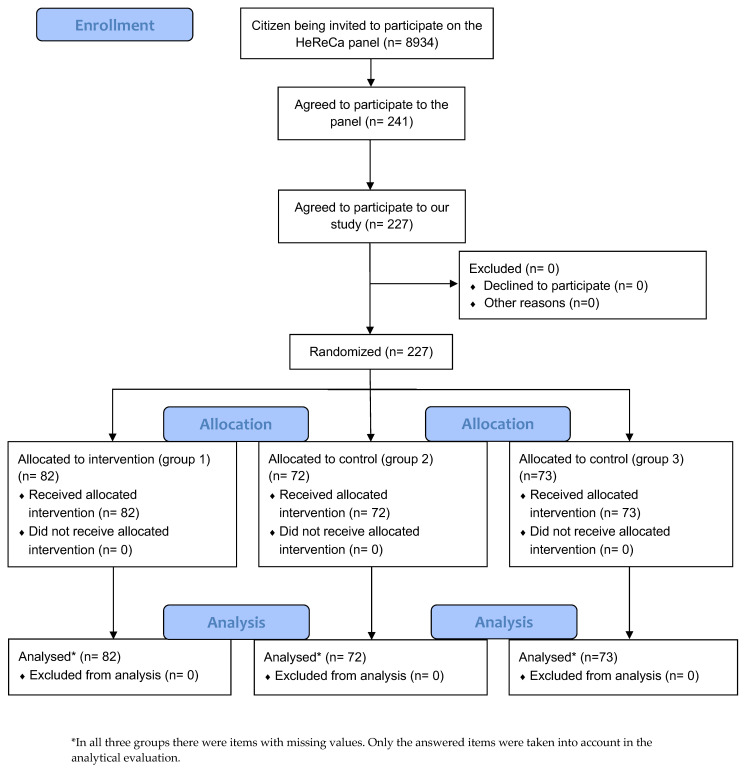
CONSORT Flow Diagram [20].

**Table 1 ijerph-20-02165-t001:** Baseline characteristics of participants (n = 227).

	Fact Box	Fact Box	Fact Box
	Natural Frequencies n *=* 82	Percentages n *=* 72	Graphic Chart n *=* 73
Age, mean (SD)	50.8 (16.9)	50.1 (16.8)	47.7 (13.3)
Sex, n (%) *			
Female	36 (43.9)	37 (51.4)	35 (47.9)
Male	28 (34.1)	24 (33.3)	21 (28.8)
Divers	1 (1.2)	0 (0)	0 (0)
Educational level, n (%) *			
Graduation **	5 (6.1)	3 (4.2)	4 (5.5)
Master (university) **	19 (23.2)	15 (20.8)	25 (34.2)
Master (university of applied sciences) **	11 (13.4)	12 (16.7)	6 (8.2)
Bachelor (university of applied sciences and university) **	7 (8.5)	5 (6.9)	5 (6.8)
Higher School Certificate (Abitur)	42 (51.2)	33 (45.8)	47 (57.5)
Higher School Certificate (qualification for a university of applied sciences)	7 (8.5)	9 (12.5)	5 (6.8)
Secondary School	12 (14.6)	12 (16.7)	7 (9.6)
Lower Secondary School	1 (1.2)	4 (5.6)	1 (1.4)
Others	1 (1.2)	2 (2.8)	1 (1.4)
Students attending general schools (full-time or part-time)	1 (1.2)	0 (0)	0 (0)
No response	18 (22.0)	12 (16.7)	17 (23.3)

* Due to missing values, numbers do not always sum up to 100%. ** Multiple answers were possible.

**Table 2 ijerph-20-02165-t002:** Primary outcome: verbatim and gist knowledge (sum score).

Outcome	Fact Box	Median	Mean (±SD ^a^)	Min-Max	95% CI ^b^	*p*-Value ^c^
Verbatim knowledge	Natural frequencies	0.78	0.63 (0.36)	0–1		
Verbatim knowledge	Percentages	0.78	0.64 (0.36)	0–1	−0.004; −0.004	0.986
Verbatim knowledge	Graphic chart	0.78	0.63 (0.36)	0–1		
Gist knowledge	Natural frequencies	0.80	0.69 (0.37)	0–1		
Gist knowledge	Percentages	0.80	0.71 (0.35)	0–1	−0.004; 0.002	0.708
Gist knowledge	Graphic chart	1.00	0.72 (0.37)	0–1		

^a^ SD = standard deviation. ^b^ 95% CIs ofdifferences of means calculated based on ANOVA. ^c^ Asymptotic significance from two-sided Kruskal–Wallis tests.

**Table 3 ijerph-20-02165-t003:** Participants’ responses regarding readability/comprehensibility, acceptance, and relevance.

Outcome	Fact Box	Mean (±SD ^a^)	Min-Max	*p*-Value ^b^
Readability/Comprehensibility	Natural frequencies	2.19 (0.62)	0–3	
Readability/Comprehensibility	Percentages	2.10 (0.57)	0–3	0.657
Readability/Comprehensibility	Graphic chart	2.14 (0.47)	1–3	
Acceptance	Natural frequencies	1.71 (0.70)	0–3	
Acceptance	Percentages	1.81 (0.57)	0–3	0.430
Acceptance	Graphic chart	1.85 (0.53)	0.33–3	
Relevance	Natural frequencies	1.94 (0.77)	0–3	
Relevance	Percentages	1.90 (0.66)	0–3	0.798
Relevance	Graphic chart	1.96 (0.74)	0–3	

^a^ SD = standard deviation. ^b^ Asymptotic significance of two-sided Kruskal–Wallis tests of differences of means.

**Table 4 ijerph-20-02165-t004:** Knowledge by educational level.

Outcome	Fact Box	Educational Level	n	Mean (±SD ^a^)	Median	Difference of Means (95% CI)	*p*-Value ^b^
Verbatim knowledge	Natural frequencies	Lower education Higher education	14 50	0.69 (0.25) 0.81 (0.22)	0.72 0.89	0.12 (−0.222; 0.000)	0.182
Verbatim knowledge	Percentages	Lower education Higher education	15 45	0.62 (0.31) 0.80 (0.23)	0.67 0.89	0.22 (−0.222; 0.000)	0.049
Verbatim knowledge	Graphic chart	Lower education Higher education	8 48	0.57 (0.37) 0.81 (0.21)	0.72 0.89	0.17 (−0.222; 0.000)	0.098
Gist knowledge	Natural frequencies	Lower education Higher education	14 50	0.81 (0.20) 0.88 (0.19)	0.80 1.0	0.2 (−0.200; 0.000)	0.337
Gist knowledge	Percentages	Lower education Higher education	15 45	0.77 (0.28) 0.86 (0.22)	0.80 1.0	0.2 (−0.200; 0.000)	0.269
Gist knowledge	Graphic chart	Lower education Higher education	8 48	0.80 (0.32) 0.90 (0.18)	1.0 1.0	0 (−0.200; 0.000)	0.770

^a^ SD = standard deviation. ^b^ Asymptotic significance from two-sided Mann–Whitney U Test of difference of means.

## Data Availability

ZENODO: https://doi.org/10.5281/zenodo.7046085.

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
