# Peer review of "Efficacy of Three Numerical Presentation Formats on Lay People’s Comprehension and Risk Perception of Fact Boxes—A Randomized Controlled Pilot Study"

_ijerph, 2023, doi:10.3390/ijerph20032165_

Round 1

Reviewer 1 Report

Well written and organized manuscript. If there is a way to be more inclusive of lower educated citizens in future iterations, that would be helpful for results, however, you do include that as a limitation.  I could not find anything that needs improvement.

Author Response

We thank the reviewer for the helpful comments.

Reviewer 2 Report

Thanks very much for the invitation to review the manuscript, entitled “Efficacy of three numerical presentation formats on lay people’s comprehension and risk perception of fact boxes – a randomized controlled pilot study”. This paper hits on an interesting research topic whether fact boxes improved risk perceptionAn online experiment was carried out with 227 participants  aimed to compare the three formats natural frequencies, percentages and bar graphs in fact boxes with regard to comprehension and risk perception in general and according to the educational background. I appreciate the efforts spent by the authors in collecting data. However, I also have the following concerns.

1. There lacks a proper literature review. This study talked a lot why this topic is important from practical perspective, but didn’t mention how this study fulfills the research gaps in previous studies. The theoretical contribution needs to be further identified. Saying: "To date, fact boxes have hardly been evaluated" (page 2, line 70) is not enough to show the theoretical contribution.

2. Figures are difficult to read, and in the case of "Antibiotics for acute bronchitis", practically illegible.

3. Table A1 can not be seen so it is hard to comment on moderating variables; on what basis they were selected. Authors should specify the methods and analyzes carried out to determine constructs in questionnaire regarding knowledge/risk perception, readability/comprehensibility, acceptance, and relevance. 

4. The Materials and methods' section should include a describtion of questionnaire used in the study, especially if the moderating variables were isolated on its basis. The detailed process of questionnaire development needs to be elaborated.

5. There are existing measurements for  such factors: “knowledge”, "readability" etc. in previous studies. I would like to know why the authors did not use existing measurements.

6. There should be more information about previous focus group study mentioned on page 8, line 354. Who took part in it, how large was the group, etc.?

7. Results. More sophisticated statistical analyzes of the obtained results should be considered.

8. The process of inclusion of participants in the study should be further explained. Limiting this to the presentation of Figure 2 seems insufficient.

9. The Discussion section is short. More research should be included to make it more developed.

10. The limitation of the insufficient representation of the elderly seems to be the main weakness of this study, as well as the lack of information about the phenomenon of "imagery of the message" in health communication and transformation of contemporary culture.

Author Response

We thank the reviewer for the helpful comments.

Comments Reviewer 2

Response

Reaction

1.

There lacks a proper literature review. This study talked a lot why this topic is important from practical perspective, but didn’t mention how this study fulfills the research gaps in previous studies. The theoretical contribution needs to be further identified. Saying: "To date, fact boxes have hardly been evaluated" (page 2, line 70) is not enough to show the theoretical contribution.

Thank you for pointing this out.

With regard to the literature search, we followed the guideline of evidence-based health information. In preparation of the preceding focus group study and this present study, we updated the literature search of the guideline of evidence-based health information.

We have added the information on the literature search in the introduction.

2.

Figures are difficult to read, and in the case of "Antibiotics for acute bronchitis", practically illegible.

Thank you for pointing this out. The submitted versions were easy to read.

We will discuss this with MDPI support.

3.

Table A1 can not be seen so it is hard to comment on moderating variables; on what basis they were selected. Authors should specify the methods and analyzes carried out to determine constructs in questionnaire regarding knowledge/risk perception, readability/comprehensibility, acceptance, and relevance.

Thank you for pointing this out.

We are sorry, but the submitted versions were easy to read.

The following explanations address the points 3-5

We chose the moderator variable education based on the literature. The panel asked about the school-leaving qualification and the educational qualification.

We followed the guideline evidence-based health information [4] in developing the questionnaire regarding established outcomes.

The questions to survey knowledge and risk perception had been adapted from previous studies. The development processes in these previous studies were based on the ethical guidelines of the General Medical Council. This reference was the first one to define what information has to be given before patients can make decisions. The questions for the secondary outcomes had been adapted from studies that were included in the guideline evidence based health information.

Finally we tested the questionnaire in a focus group study. In an iterative process we optimized the wording until the questions were well understood.

We will discuss this with MDPI support.

We have added the explanations regarding the development and piloting of the questionnaire in the manuscript and also added further relevant references in the materials and methods section (2.6).

4.

The Materials and methods' section should include a describtion of questionnaire used in the study, especially if the moderating variables were isolated on its basis. The detailed process of questionnaire development needs to be elaborated.

We refer to our answer in point 3.

5.

There are existing measurements for such factors: “knowledge”, "readability" etc. in previous studies. I would like to know why the authors did not use existing measurements.

We refer to our answer in point 3.

6.

There should be more information about previous focus group study mentioned on page 8, line 354. Who took part in it, how large was the group, etc.?

We apologise, this note is unclear. Perhaps the page references in the reviewers' document differ? A focus group study has been conducted and reported the development process of the fact boxes and the evaluation in focus groups. Due to lack of space, we refer to the reference of the focus group study.

Aubertin P, Hinneburg J, Hille L, Steckelberg A. Fact Boxes: what gets through? A focus group study, Z Evid Fortbild Qual Gesundhwes. 2022 Feb 8:S1865-9217(22)00004-6. doi: 10.1016/j.zefq.2021.12.011.

7.

Results. More sophisticated statistical analyzes of the obtained results should be considered.

We believe that the analytical methods selected are the right ones. The analyses were performed according to the registered study protocol.

8.

The process of inclusion of participants in the study should be further explained. Limiting this to the presentation of Figure 2 seems insufficient.

Thank you very much for this comment

We have reformulated the recruitment process. It should be more understandable now.

9.

The Discussion section is short. More research should be included to make it more developed.

Many thanks for the comment.

We have added further references to the discussion.

10.

The limitation of the insufficient representation of the elderly seems to be the main weakness of this study, as well as the lack of information about the phenomenon of "imagery of the message" in health communication and transformation of contemporary culture.

Thank you for the comment.

We rather see the problem that the weak educational strata were underrepresented. We addressed that in the discussion.

Your second comment was not that clear for us and rather a general comment. We hope that the overall revision also addressed this aspect.

Reviewer 3 Report

This paper discusses how different forms of a fact box influence people's knowledge, comprehensibility, acceptance, and perceived relevance of certain medications. Specifically, the current research compares three different formats of a fact box—that is, presenting treatment effects as natural frequencies, percentages, and via graphic charts. Conducting a randomized controlled study, the authors demonstrate the efficacy of the three formats of a fact box. While the research question is both theoretically and practically important, I believe there are a variety of ways in which the paper could be strengthened, which I will outline below.

Introduction

(1) The authors start the introduction by showing that there is a lack of evidence-based health information. However, the authors do not specify what evidence-based health information is and what kind of evidence-based health information lacks. I suggest the authors to use an example to clarify the situations where evidence-based health information is used and why presenting it in a readily comprehensible way is practically important.

(2) The introduction is less straightforward. The authors had better not talk so much about the other ways to communicate evidence-based health information, which are less relevant to the research question. Since the focus of this research is different forms of a fact box rather than the comparison between a fact box and other ways to present evidence-based health information, the authors should revise the introduction section to make the research question more straightforward and compelling.

Methods

(1) There are three conditions: 1) the fact box presents treatment effects as natural frequencies, 2) the fact box uses percentages, and 3) the fact box uses a graphical presentation. However, figure 1 does not specify each condition. The authors should make the materials for each condition more clearly.

(2) The authors treat knowledge and risk perception as one outcome variable. However, theoretically, knowledge about something is conceptually different from risk perception about something. To illustrate, one's knowledge about the risk of antibiotic treatment can lead to an increase in his/her risk perception. Given that knowledge and risk perception are divergent concepts, the author should clearly define the outcome variable and use different items to measure different constructs.

(3) 8,934 participants were invited to join the study, but only 227 participants agree to participate in the survey. The non-response bias could be a big concern. To assess the non-response bias, the author should randomly select several participants who did not participate in the survey and ask questions about some key variables in order to demonstrate there is no significant differences between participants and non-participants.

(4) Why do the authors label the study as a pilot study?  Usually, when we use a pilot study, a formal study would follow. Since the research has only one study, I am confused by why the authors call it a pilot study

Theoretical Contributions

One of the contributions of the current research would be the different effects of the three formats of fact boxes. However, most of the effects of the three formats on outcome variables are not significant. Given that in most cases, there is no difference between the three fact box formats, why would the authors classify and differentiate the three formats?

Writing

I understand that English is not your native language (and I can relate). However, the writing is often below IJERPH standard. I would suggest soliciting the services of a professional editor. 

Author Response

We thank the reviewer for the helpful comments.

Comments Reviewer 3

Response

Reaction

1.

The authors start the introduction by showing that there is a lack of evidence-based health information. However, the authors do not specify what evidence-based health information is and what kind of evidence-based health information lacks. I suggest the authors to use an example to clarify the situations where evidence-based health information is used and why presenting it in a readily comprehensible way is practically important.

Many thanks for the comment.

We have added further information on evidence based information but did not insert further examples since the fact boxes are an example.

2.

The introduction is less straightforward. The authors had better not talk so much about the other ways to communicate evidence-based health information, which are less relevant to the research question. Since the focus of this research is different forms of a fact box rather than the comparison between a fact box and other ways to present evidence-based health information, the authors should revise the introduction section to make the research question more straightforward and compelling.

We thank you for the comment.

We have revised the introduction.

Methods

1.

There are three conditions: 1) the fact box presents treatment effects as natural frequencies, 2) the fact box uses percentages, and 3) the fact box uses a graphical presentation. However, figure 1 does not specify each condition. The authors should make the materials for each condition more clearly.

Thank you for pointing this out and we apologize for the technical problems. The submitted inserted figures were complete in our case.

We now present figures 1-3 for each presentation format. The technical problems will be discussed with the MDPI support.

2.

The authors treat knowledge and risk perception as one outcome variable. However, theoretically, knowledge about something is conceptually different from risk perception about something. To illustrate, one's knowledge about the risk of antibiotic treatment can lead to an increase in his/her risk perception. Given that knowledge and risk perception are divergent concepts, the author should clearly define the outcome variable and use different items to measure different constructs.

Thank you very much. We do understand the critique you raised.

In the field of health information, the outcome risk perception is widely used. However, it should rather be risk assessment, since it mostly addresses that probabilities of risks are understood. We referred to the outcomes that had been included in the guideline evidence based health information.

We now consequently changed the wording to verbatim and gist knowledge.

3.

8,934 participants were invited to join the study, but only 227 participants agree to participate in the survey. The non-response bias could be a big concern. To assess the non-response bias, the author should randomly select several participants who did not participate in the survey and ask questions about some key variables in order to demonstrate there is no significant differences between participants and non-participants.

We can understand that this is confusing.

We revised the recruitment section and separated the recruitment for the panel from the recruitment for this study. Actually 241 people were invited to take part in this study and 227 agreed to the invitation.

4.

Why do the authors label the study as a pilot study?  Usually, when we use a pilot study, a formal study would follow. Since the research has only one study, I am confused by why the authors call it a pilot study.

Thank you for pointing this out.

We called this study a pilot study, because we wanted to determine the feasibility and acceptability of the procedures and also gain information for further sample size calculations for a main randomized controlled trial (RCT)”.

Theoretical Contributions

One of the contributions of the current research would be the different effects of the three formats of fact boxes. However, most of the effects of the three formats on outcome variables are not significant. Given that in most cases, there is no difference between the three fact box formats, why would the authors classify and differentiate the three formats?

Thank you for this note.

According to the literature, we generated our hypotheses and conducted this pilot study. As outlined in the previous comment we would now refrain from conducting further RCTs to test the hypotheses in larger samples but rather change the focus.

However, since we did not find any differences the results rather suggest that further studies should take the individual preferences into account. We added further information to the discussion.

Writing

I understand that English is not your native language (and I can relate). However, the writing is often below IJERPH standard. I would suggest soliciting the services of a professional editor.

We followed your suggestion.

The manuscript was carefully checked by a native speaker.

Round 2

Reviewer 2 Report

The corrections introduced by the authors seem to be sufficient and satisfactory.

I have no comments on the current version of the manuscript, and recommend the publication in its current form.